# Screening and Characterization of a New Iflavirus Virus in the Fruit Tree Pest *Pyrops candelaria*

**DOI:** 10.3390/insects15080625

**Published:** 2024-08-19

**Authors:** Hong Lin, Weitao Song, Dongmei Ma, Chi Yang, Yanfang Yao, Renyi Liu, Ling Hao, Dandan Wu, Shihua Wang, Jimou Jiang, Jun Xiong, Rui Ma, Jiajing Xiao, Zhenhong Zhuang

**Affiliations:** 1Key Laboratory of Pathogenic Fungi and Mycotoxins of Fujian Province, Key Laboratory of Biopesticide and Chemical Biology of Education Ministry, Proteomic Research Center, and School of Life Sciences, Fujian Agriculture and Forestry University, Fuzhou 350002, China; linhong2805@163.com (H.L.); swtwzsz@163.com (W.S.); yc113078@163.com (C.Y.); 5220543002@fafu.edu (Y.Y.); hl950331@163.com (L.H.); kikimra@163.com (D.W.); wshyyl@sina.com (S.W.); xj_x0347@163.com (J.X.); 2College of Animal Sciences, Fujian Agriculture and Forestry University, Fuzhou 350002, China; 13945932275@163.com; 3Center for Agroforestry Mega Data Science, Haixia Institute of Science and Technology, Fujian Agriculture and Forestry University, Fuzhou 350002, China; ryliu@fafu.edu.cn; 4Fruit Research Institute, Fujian Academy of Agricultural Science, Fuzhou 350013, China; jjm2516@126.com; 5College of Resources and Environment, Fujian Agriculture and Forestry University, Fuzhou 350002, China; 13199871065@163.com

**Keywords:** iflavirus, *Pyrops candelaria*, virus, insect, distribution

## Abstract

**Simple Summary:**

The homopteran insect, *Pyrops candelaria* (longan lanternfly), mainly harms longan trees, lychees, olives and other crops by absorbing the sap on the trunk. In this study, the reverse etiology method was used to find the pathogens carried by *P. candelaria*. Through RNA–seq, we found a new iflavirus in *P. candelaria*, which was first reported in *P. candelaria*, and we named it PyCaV (*Pyrops candelaria* associated virus). Identified by a pair of specific primers, it is worth noting that the presence of PyCaV can be identified in the head, chest and abdomen of *P. candelaria*, and that PyCaV infection rate of *P. candelaria* is affected by time and location. The above results provide new insights to reveal the role of *P. candelaria* as a potential vehicle for microbial pathogens and extend our understanding to iflavirus.

**Abstract:**

*Pyrops candelaria* is one of the common pests of fruit trees, but the research on the pathogenic microorganisms it may carry is very limited. Therefore, it is essential to reveal the pathogenic microbes it carries and their potential hazards. This study found a new virus from the transcriptome of *P. candelaria*, which was first reported in *P. candelaria* and named PyCaV (*Pyrops candelaria* associated virus). RACE and bioinformatics assay revealed that the full length of PyCaV is 10,855 bp with the polyA tail, containing a single open-reading frame (ORF) encoding a polyprotein consisting of 3171 amino acid (aa). The virus has a typical iflavirus structure, including two *rhv* domains, an RNA helicase domain (HEL), a 3C cysteine protease domain (Pro), and an RNA–dependent RNA polymerase domain (RdRp). Further phylogenetic analysis revealed that this virus belongs to family *Iflaviridae* and sequence alignments analysis suggested PyCaV is a new member in an unassigned genus of family *Iflaviridae*. Further in-depth analysis of the virus infection showed that PyCaV is distributed throughout the whole *P. candelaria*, including its head, chest, and abdomen, but more PyCaV was identified in the chest. The distribution of PyCaV in different parts of *P. candelaria* was further explored, which showed that more PyCaV was detected in its piercing–sucking mouthparts and chest viscera. Statistical analysis showed that the PyCaV infection was affected by time and location.

## 1. Introduction

According to the different virus hosts, viruses can be roughly classified into plant viruses, animal viruses, and bacterial viruses. *Iflavirus*, belonging to the family *Ifaviridae*, is an animal virus composed of single-stranded, sense, non-segmented RNA sequence [1]. According to the *Iflaviridae* chapter of the ICTV online report, there are 16 recognized species within the genus *Iflavirus*. The virus particles of this genus are roughly spherical with a diameter of about 22–30 nm. Iflaviruses have the basic characteristics of small RNA viruses: the 5′ end of the RNA strand is connected to the viral protein genome (VPg), and the internal ribosome entry site (IRES)a is contained in the 5′ non-coding region, which mediates the initiation of RNA translation [2,3]. Therefore, iflaviruses can initiate RNA translation independent of the 5′ cap structure [4]. Iflaviruses have a single open-reading frame (ORF), which can be directly translated into a polyprotein, which then post-translationally hydrolyzed into several structural and nonstructural proteins (functional proteins); i.e., structural proteins and functional proteins are cleaved and separated after translation before implementing their biological functions. The 3′ non-coding region of iflaviruses contains a typical poly-A (adenine) tail [2,5]. According to the related reports, a variety of iflaviruses have been found, such as deformed wing virus [6], infectious flacheric virus [7], *Perina nuda picornalike* virus [8], *Ectropis obliqua* picornalike virus [9], *Varrora destructor* virus-1 virus [10] and *sacbrood* virus (SBV) [11].

At present, most known hosts of iflaviruses are arthropods, especially members of Hemiptera, Lepidoptera, Hymenoptera, Diptera, Coleoptera and Orthoptera [12,13,14]. Infested hosts exhibit abnormal growth and development and have a high mortality rate [15]. However, not all iflaviruses cause observable pathology, and many appear to be transmitted vertically. The gregariousness of insects promotes the spread of virus in their community [16]. Once the iflaviruses enter the host cell, the infection process is very rapid, and the offsprings of the virus replicate in large numbers within a few hours. When iflaviruses infect the host, it will quickly infect the brain, intestine, and gland tissues of the host [17]. Iflaviruses pose a significant threat to certain essential environmental insects. It has been reported that the iflaviruses infection of *Spodoptera exigua* can increase the host’s susceptibility to nucleopolyhedrovirus infection, which could increase the larval mortality of *S. exigua* [18]. Infected by iflaviruses, the weight of *S. exigua* larvae is reduced and the sex ratio is unbalanced. *Helicoverpa armigera* is one of the most important agricultural pests of cotton and other important economic crops in the world. Iflaviruses are mainly distributed in the fat bodies of larvae and adults of the infected *H. armigera*, which is not conducive to the growth and reproduction of this notorious pest [19]. In some cases, iflaviruses infection can be fatal to the host. SBV mainly exists in the fat cells of the larvae of honeybees, impairing the metabolic function of honeybees, leading the failure of bee pupation, and ultimately causing the death of the hosts [20,21].

The homopteran insect, *Pyrops candelaria,* is strange and novel for its long nose and brightly colorful wings. *P. candelaria* is also called longan lanternfly, mainly damaging longan, litchi, olive, orange, mulberry, mango, and pomelo crops by sucking the juice from the trunks of these fruit trees [22,23]. Longan lanternflies prefer warm temperature and high moisture, so they are distributed in the fruit-bearing forests in the low-altitude regions of the tropic and subtropic area, and they are found mainly in China (including Hainan, Fujian, Guangdong, Guangxi, and Hong Kong), Thailand, Vietnam, Malaysia, Cambodia, Sikkim, India, and Indonesia [22,23]. The development of *P. candelaria* experiences three developmental stages: egg, nymph, and adult. Their host preference is broad in the nymph stage but narrows to a few plant species in the adult stage, and the invasion of longan lanternflies serious endangers the culture and harvest of pomelo and longan trees [23]. The nymph of *P. candelaria* is good at bouncing, and the adult is good at bouncing and flying [24]. *P. candelaria* mainly endangers longan trees; when *P. candelaria* is prevalent, the dense population of the insects causes serious damage to longan trees. Adults and nymphs suck plant juice by inserting their needle-like mouthparts into the cortex of trunks and branches. After being sucked by longan lanternflies, small black spots would gradually appear in the cortex [25]. When the population of *P. candelaria* becomes too large, the harm caused by them might become serious, which can weaken the longan trees, stunt their growth, make branches dry, and result in fruit drop or poor fruit quality, and the excreta of this insect can even cause soot disease of longan tree [25]. Researchers found that in a forest mixed planting of longan, litchi, olive and mango, the population of this insect is usually especially large [26].

Longan lanternflies have co-evolved with various fruit trees, microbial pathogens, and other environmental insects through long-term interaction, and they may play an important role in the transmission of some microbial pathogens of fruit trees or from other agricultural important insects. With the advancement of sequencing technology, microbial research has also entered the era of omics, in which to screen and identify possible microbial pathogens is one of the most important parts. The study found a new virus in the course of screening microbial pathogens carried by *P. candelaria* through high-throughput RNA sequencing, and the virus was further characterized in order to lay a solid foundation for revealing the possible interaction of the virus and its hosts, providing new insights to reveal the role of *P. candelaria* as a potential vector for microbial pathogens.

## 2. Materials and Methods 

### 2.1. Insect Sample Collection

*P. candelaria* samples were collected from the longan forest in Fujian Academy of Agricultural Sciences (FRIFAAS) and Minhou County.

### 2.2. RNA Sequencing (RNA–Seq) and Data Assembly

In this study, the collected *P. candelaria* samples were frozen and transported to BGI (Beijing Genomics Institute) for RNA extraction and RNA–seq. In this study, samples were collected for deep RNA–seq. After the possible contaminated DNA was degraded by DNaseI, ribosomal RNA was removed from the total RNA with the kit of Ribo-Zero^TM^, and all mRNA was retained to the greatest extent. NEB Fragmentation Buffer was mixed with the greatest extent of retained mRNA to cut the RNA into 250 to 300 bp short fragments. Then, the first strand of cDNA was synthesized with random hexamers as primers, and the second strand of cDNA was further synthesized by adding buffer, dNTPs, RNase H, and DNA polymerase I at Thermomixer, and then it was purified by AMPure XP beads. A-Tailing Mix was added to repair the end of double-stranded cDNA and add an A base at the 3′ end of it. Then, the linker mix was added to connect the linker with cDNA; the joint product was purified and recovered with magnetic beads. The library was established by amplifying recovered cDNA in a PCR mix, the concentration of the library was quantified by Qubit2.0 Fluorometer, and the library was diluted to 1.5 ng/µL. At the same time, the quality of the library was assayed by Agilent 2100 Bioanalyzer (Agilent, Wilmington, USA). The qualified library is denatured into a single chain by adding NaOH, and the denatured and diluted library was added into FlowCell, hybridized with the linker in the FlowCell, and completed bridge PCR amplification in cBot. The sequencing platform was illumina novaseq6000 to generate 15.48 GB 150 bp paired-end reads. In quality control, the software fastqc (version 0.11.8) was chosen to check the raw clean reads from RNA–seq, and then the quality of clean reads was checked by software fastp (version 0.20.1) by which low-quality sequence reads were removed again. Following quality control, the clean reads were assembled de novo by Trinity (version 2.15.1) with default assembly parameters and blasted in the database of the virus (https://ftp.ncbi.nlm.nih.gov/refseq/release/viral/, accessed on 16 February 2024). 

### 2.3. Rapid Amplification of cDNA Ends

RNA was extracted from *P. candelaria* samples according to the methods described in Section 2.2, and cDNA was obtained by reverse transcription. The specific primers were used to identify whether the sample carried PyCaV virus, and the cDNA carrying the virus was stored at −20 °C. Race double-end specific primers and nested primers were designed according to the sequence obtained by the second RNA–seq carried out in this study. PCR amplification was performed using cDNA as a template using 5′-RACE end-specific primers and reverse transcription primers, and the PCR products were detected by agarose gel electrophoresis to ensure specific amplification. The PCR product was purified, and the purified cDNA chain was used as a template for subsequent experiments. The adaptor was ligated to the 5′ end of the cDNA chain. After the ligation reaction was completed, the adaptor-specific primers and internal primers were used for nested PCR amplification. The first round of PCR amplification was performed using internal primers and adapter-specific primers. The first round of PCR product was used as a template, and the second round of PCR amplification was performed using nested primers. PCR products were detected by agarose gel electrophoresis to ensure specific amplification. The PCR product was transformed into an *E. coli* Competent Cell JM109 vector, and the positive clones were selected for sequencing to obtain the 5′ and 3′ end information of the cDNA sequence. The 3′RACE amplification primers of the target gene were designed, and the obtained cDNA was used as a template for 3′RACE PCR amplification. The PCR product was purified and recovered by DNA, and the 3-terminal full-length sequence was obtained by DNA sequencing. 

### 2.4. Bioinformatics Analysis

The conserved domain of the amino acid sequence was predicted with the NCBI Conserved Domain Database (CDD, https://www.ncbi.nlm.nih.gov/Structure/cdd/wrpsb.cgi, accessed on 16 December 2023). Phylogenetic analysis was performed based on the aligned conserved RdRp domain sequence from PyCaV and other members in family *Iflaviridae*; two members in *Picornavirales* were selected as an outgroup. MAFFT (v7.310) [27] was used to perform multiple sequence alignment with default parameters, while the best substitution model was estimated by Prottest3 (version 3.4.2) [28]. A maximum likelihood phylogenetic tree was constructed using PhyML (version 3.2.0) with 1000 bootstrap replicates with the best estimated model [29].

### 2.5. RNA Extraction

In this study, the collected *P. candelaria* samples were subjected to independent RNA extraction after alcohol disinfection. The preparation of RNA was following the protocol previously used by Toni et al. with minor modifications [30]. The insect samples were first disinfected with sodium hypochlorite and ground into powder by liquid nitrogen. After 1 mL Trizol reagent was mixed with the powder for 30 min at 4 °C, 200 μL of pre-cooling dichloromethane was added and mixed thoroughly for 2 min, after which it was centrifuged at 4 °C, 12,000 r/min for 20 min. A portion of 450 μL supernatant was transferred into the new sterile RNase EP tube and mixed with another 200 μL of dichloromethane, mixed repeatedly for 2 min, and then centrifuged at 4 °C at 12,000 r/min for 10 min. Following, 350 μL of supernatant was transferred into a new sterile RNase EP tube; the same amount of pre-cooling isopropanol was added and mixed, and it was kept under −20 °C for 20 min, after which it was centrifuged under 4 °C at 12,000 r/min for 20 min. The pellets were washed with 1 mL of pre-cooled 75% ethanol (with DEPC water) twice, air dried and finally dissolved in 20 μL DEPC water. The quality of prepared RNA samples was monitored by electrophoresis and with Nanodrop 2000, and the qualified RNA samples (the ratio of OD260/OD280 reaches about 2.0) were stored at −80 °C for further use.

### 2.6. cDNA Preparation

The cDNA was obtained by the reverse transcription of total RNA using a Revert Aid First-Strand cDNA Synthesis Kit (Quan, Beijing, China). First, 5 μg of RNA, 1 μL of Oligo [31] 18 Prime, and 2 μL of RNase–free water supplement were mixed and put in a 65 °C water bath for 5 min. Then, they were transferred immediately into an ice bath for 2 min. After the ice bath, 10 μL of R-Mix, 1 μL of E-Mix, and 1 μL of gDNA-Remover were added to the system and kept at 42 °C for 30 min, after which they were heat-shocked at 85 °C for 5 s to inactivate the activity of the reverse transcriptase. Finally, the prepared cDNA was stored at −20 °C.

### 2.7. DNA Purification

The PyCaV virus fragment was amplified by PCR with specific primers, the DNA was purified and recovered, and the recovered DNA fragment was sequenced. After electrophoresis, a gel recovery kit (Gene Gel Extraction Kit (Omega, Shanghai, China)) was used for the recovery of DNA products. The amplified DNA fragment was mixed with a 10×DNA loading buffer and analyzed by electrophoresis with a 1% agarose gel at 200 V (180 mA) in TBE buffer for 20 min. After electrophoresis, the target band was cut out and transferred into a sterile EP tube and mixed with the same volume of binding buffer; then, the gel was melted in a 65 °C water bath. The melted mixture was transferred to an adsorption column from the gel recovery kit and centrifuged at 10,000 r/min for 1 min, after which the filtrate was discarded. Following, 300 μL of binding buffer was added to the adsorption column, and it was centrifuged at 12,000 r/min for 1 min; then, 700 μL of SPW wash buffer was added to the adsorption column, and it was centrifuged at 14,000 r/min for 1 min. The adsorption column was centrifuged at 15,000 r/min for 2 more min, which was followed by adding 30 μL of elution buffer to the adsorption column. After standing at room temperature for 2 min, it was then centrifuged at 15,000 r/min for 2 min, and the DNA recovery product was collected. Finally, the concentration of DNA recovery product was measured by Nanodrop 2000 (Thermo Scientific, Wilmington, USA) and stored at −20 °C for further use.

### 2.8. Virus Verification

In this study, the following methods were used to identify whether the *P. candelaria* samples carried the PyCaV virus. After the surface was disinfected by sodium hypochlorite, the *P. candelaria* samples were ground into a powder with liquid nitrogen and lysed at 4 °C for 30 min with 1 mL of Trizol reagent, which was followed by adding 500 µL chloroform and standing for 3 min. Then, the samples were centrifuged at 12,000 rpm for 1 min. After the supernatant was mixed with the same volume of isopropanol and kept still at −20 °C for 30 min, the mixture was centrifuged at 12,000 rpm for 1 min. The supernatant was discarded, and the pellet was thoroughly washed with 20% ethanol. After being further centrifuged at 12,000 rpm for 1 min, the pellet was left standing to air dry for 10 min. Finally, the prepared total RNA was dissolved in 20 µL DEPC pre-treated water and kept at −80 °C for further use. The total RNA was reverse transcribed into cDNA with the TransScript^®^One-Step gDNA Removal and cDNA Synthesis SuperMix kit (Transgen Biotech, Beijing, China). The cDNA was used as a template and amplified with primers designed according to the sequence information obtained from RNA–seq, and the amplification results were finally verified by agarose gel electrophoresis and sequenced by Fuzhou Sunya Biotechnology Co., Ltd. (Fuzhou, China).

### 2.9. Statistics Analysis and Model Construction

The samples of *P. candelaria* were collected and classified according to the location and time, and they were collected from two locations: the Fruit Research Institute of Fujian Academy of Agricultural Science (FRIFAAS) and Minhou County of Fuzhou City. The total sample number and number of samples carrying viruses at each time were counted, and the positive rate was calculated (Number of samples carrying viruses/The total sample number). The histogram was established by importing statistical data into the software GraphPad Prism 9.0, and a *t*-test was selected for significance analysis (* *p* < 0.05). The simulated model was constructed with Origin 2019 via the method of the scatter plot, and related curve trends were selected according to the distribution pattern of the obtained points. Finally, a nonlinear curve was chosen, and the logistic model was selected for further fitting.

## 3. Results

### 3.1. The Virus Screening from P. candelaria by RNA–Seq

To screen the microbial pathogens carried by the longan lanternflies, an RNA–seq analysis was carried out. A total of 51.61 million paired-end reads were generated and then assembled de novo by the software Trinity (version 2.15.1), generating 1,117,086 candidate contigs for further analysis. To investigate the sequences originated from viruses, the 372,784 contigs with length no less than 500 nt were used to search against the viral refseq proteins from NCBI through BLASTX, and 7658 contigs showed best hits against viral proteins longer than 200 aa. Among these candidate viral-derived contigs, only 1932 were non-redundant contigs from 293 kinds of viruses with the phage removed (Appendix A). Based on the classification information, viral contigs from 61 families were detected in *P. candelaria* (Figure 1). Among them, about 217 kinds of viruses detected in *P. candelaria* belong to dsDNA, 18 kinds belong to ssDNA(+), 10 kinds belong to ssRNA (+/), and 10 belong to ssRNA–RT.

The longest hit length blasted with known virus was found to be a 10,916 nucleotide (nt) contig belonging to ssRNA (+) virus, of which 37.27% of its identity was shared with the polyprotein of *Nilaparvata lugens honeydew virus-3* (NLHV3), which is a member of the family *Iflaviridae* (Appendix A). Another contig of 6920 nt shared 23.18% identity with RdRp of Culex phasma-like virus, which is a member of the family Phasmaviridae. The limited sequence identity between the viral contig and other known viruses suggested the presence of several new viruses carried by *P. candelaria*. According to the above results and considering the length and the identity of these two contigs, the longest contig (10,916 nt) of the viral ssRNA (+) was selected for further analysis in this study, and the potential ssRNA (+) virus from *P. candelaria* was tentatively named PyCaV (*Pyrops candelaria* associated virus) in the following sections.

### 3.2. The Full-Length Sequencing and Related Bioinformatics Analysis Reflect PyCaV a New Virus

In order to obtain the full length of the PyCaV sequence, Rapid Amplification cDNA End (RACE) experiments were carried out based on the assembled contig of 10,916 nt and the primers used in this study accessed in Appendix A. The sequence of PyCaV was validated by Sanger Sequencing, and the final full sequence of the virus was submitted to NCBI GenBank (ID: ON382046). With a polyA tail, the full-genome sequence of the PyCaV virus is 10,855 bp. Similar to other members of the family *Iflaviridae*, PyCaV contains only a single predicted open-reading frame (ORF), and the ORF encodes a 3171 aa polyprotein (Figure 2A). The polyprotein encoded by the PyCaV virus can be cut into structural proteins and non-structural proteins. The conserved protein domains of the PyCaV virus were predicted using the NCBI conserved domain database, including two *rhv* domains, one RNA helicase domain (HEL), one 3C cysteine protease domain (Pro) and one RNA–dependent RNA polymerase domain (RdRp) (Figure 2A). The results showed that the PyCaV virus had a typical iflavirus structure. When mapping the RNA–seq reads from *P. candelaria* samples to PyCaV, a total of 12,506 reads were found mapped to the reference genome (Figure 2B), suggesting the high abundance of PyCaV virus in *P. candelaria*.

### 3.3. Construction of PyCaV Phylogenetic Tree

In order to explore the phylogenetic relationship between PyCaV and other viral members in *Iflaviridae*, two members of the small RNA viruses *Rabovirus D1* (RVD) and *Aimelvirus 1* (AV1) were used as out groups. The phylogenetic tree was constructed using the conserved RdRp domain sequence of the PyCaV virus and the sequence of represented members from the family *Iflaviridae* (Figure 2C). Phylogenetic analysis showed that PyCaV was closely related to NLHV3. Further sequence alignments with the structure protein of PyCaV and other represented members reflected limited sequence identity between PyCaV and other known members of *Iflaviridae*, while the structure protein of PyCaV shared the highest sequence identity (67.67%) with that of NLHV3. Based on the genus demarcation criteria of family *Iflaviridae* from ICTV, the sequence identity at the amino acid level between the capsid proteins of PyCaV and other reported members is lower than 90%, suggesting that PyCaV is a new unclassified member of family *Iflaviridae* carried by *P. candelaria*.

### 3.4. P. candelaria Is One of the Important Carriers of the Virus PyCaV

In order to facilitate the subsequent identification of PyCaV in *P. candelaria*, we designed a pair of specific primers from the 4646 nt contig to identify PyCaV (the primers: LYJ-H-F: 5′GGTCTATGCTGTATCCAAA 3′; LYJ-H-R: 5′ATATTGTCAAGCTGGTGAG 3′), and the length of the amplified DNA fragment by the diagnostic PCR is 405 bp (Appendix A). To explore the distribution of the virus PyCaV in time, space, and inside the insect, the total RNA was extracted from the *P. candelaria* samples and reversely transcribed into cDNA. With the cDNA as templates, the results of the following diagnostic PCR showed that the virus PyCaV was detected in the two samples (Figure 3A, Lane 3 and 4) from the eight randomly collected *P. candelaria* samples from FRIFAAS (Fruit Research Institute of Fujian Academy of Agricultural Science, the green spot in the Figure 3B) and Minhou County (the red spot in the Figure 3B). The amplified DNA fragment was further collected, purified and confirmed by DNA sequencing by Fuzhou Sunya Biotechnology Co., Ltd. (Appendix A). In order to detect the distribution of the virus PyCaV in various parts of the insect, the cDNA was prepared from three parts of the *P. candelaria*: head, chest, and abdomen (Figure 3C), and the diagnostic PCR result showed that the virus PyCaV can infect the whole body of the insect, including its head, chest, and abdomen, and there is more virus found in its chest (Figure 3D, the left panel Group 1: Lane 1 to 3). In order to study the distribution of the PyCaV virus inside its body, the piercing–sucking mouthparts (PSMs), chest viscera (CV), and abdomen viscera (AV) are separated under a dissecting microscope (Figure 3C), and the PCR result showed that most PyCaV virus are located in its PSM and CV (Figure 3D, the left panel Group 2: Lane 1 to 3), and what is interesting is that the main tissue in CV is salivary glands.

To explore if the location or distance could affect the distribution of the PyCaV, the PyCaV carrying rate between the *P. candelaria* samples from FRIFAAS and Minhou County was compared by one-way ANOVA, and the result reflects that there is a significant difference in the distribution of the virus between the two sampling sites (Figure 4A and Appendix A). To analyze the possible interaction of places and seasons of the PyCaV carrying rate, 123 longan lanternflies from these two places in different seasons were collected, and it was found that 33 samples carried PyCaV. The changing trend of the PyCaV carrying rate was further simulated with Origin 2018, and the result showed the model of the PyCaV carrying rate of *P. candelaria* samples from Minhou County in different time points from spring to winter fitting to the nonlinear curve (Adj-r^2^ = 0.97184), which reflected that from summer to winter and spring, the PyCaV carrying rate is stably increased (Figure 4B). However, the carrying rate model of the samples from FRIFAAS are not fitting to the nonlinear curve or generalized linear model; the fitted model showed that the changing trend of PyCaV is wavy, in which it reaches its peak in autumn and spring, and it touches its valley in summer and winter (Figure 4B). These results reflected that the PyCaV carrying rates in these two sampling places are showing obviously different changing trends in the time dimension.

## 4. Discussion

Through RNA–seq, based on the classification information, viral contigs from 57 families and four unclassified groups were detected in *P. candelaria* (Figure 1), which broadened our understanding of the virus that can be carried by *P. candelaria*. Apart from PyCaV, a near full-length contig showed limited sequence identified with a member in Phasmaviridae, suggesting the high possibility of another virus carried by *P. candelaria*. Through RACE experiments and the construction of related phylogenetic trees, it was found that the similarity between PyCaV and NLHV3 was 67.67%, and it was distributed in the same cluster on the phylogenetic tree (Figure 2C). Based on the species demarcation criteria from ICTV, our results supported that PyCaV is a new member of the family *Iflaviridae* carried by *P. candelaria*. Since iflaviruses only infect arthropods, especially insects, and as the largest group of animal, insects account for two thirds of animal species [32]. In addition, insects occupy most of the ecological niches. More than 80% of the flowering plants in the world are considered to be dependent on insect pollination, and about three quarters of crops are also dependent on insect pollination [33,34]. Insects also become an important part of modern terrestrial ecosystems by regulating the carbon cycle in ecosystems [35,36]. Therefore, it is particularly important to further reveal the relationship between iflaviruses and insect hosts.

*P. candelaria* is not only an important pest in agriculture but also an important medium for transmitting microbial pathogens. Iflaviruses impose different biological effects after infecting the host, which is manifested as abnormal growth and development, and is even deadly to insect hosts. In recent years, some new iflaviruses have been identified. For example, Wang et al. identified a new iflavirus in *Lycorma delicatula* [37]. Chen et al. identified a new iflavirus named RdIV2 in the hemipteran insect *Recilia dorsalis*. *R. dorsalis* samples were collected in rice fields generally infected with RdIV2 and RSMV (a well-known rice-infecting virus with *R. dorsalis* as its vector), suggesting RdIV2s might affect the pathogenicity of the host by co-infection with RSMV [38]. In this study, a new iflavirus was identified from the homopteran insect *P. candelaria* and named PyCaV. This is the first time that a new iflavirus has been identified in this insect, which increases our understanding of the iflavirus and its insect host.

At present, new viruses account for almost 50% of emerging infectious diseases, which seriously threaten agricultural production, national biosafety, and even human survival [39]. Therefore, this study further explored the distribution of the PyCaV virus in the insect. It is revealed that the PyCaV virus is distributed in every part of *P. candelaria*, including the head, chest, and abdomen, which means that *P. candelaria* is, at least, one of the most important carriers of the PyCaV virus (Figure 3D, Left panel, Group 1). The detection of the body section distribution of this virus reflected that more PyCaV virus accumulated in the chest of *P. candelaria* (Figure 3D, Group 1). The chest of the insect mainly includes salivary glands and muscles, and when detecting the tissues inside the insect, it was found that the PyCaV virus mainly accumulated inside piercing–sucking mouthparts and viscera (mainly salivary glands) in the chest (Figure 3D, Group 2). The accumulation of the PyCaV virus in the piercing–sucking mouthparts and chest viscera inferred that the feeding system of the insect might play a critical role in the keeping and transmission of the virus. The salivary gland is one of the main tissues in psyllid (*Diaphorina citri*) that carries the DcFLV (*Diaphorina citri flavi*-like virus) [40]. Through the piercing–sucking mouthpart, the virions of PLRV (Potato leafroll virus) enter a green peach aphid (*Myzus persicae*) and reach the salivary glands through the circulation in the hemolymph, and the virions are released into the saliva to contaminate other fruit plants [41]. And in the small brown planthopper (*Laodelphax striatellus*), a plasma membrane-associated protein, importin α, is utilized by the RSV (rice stripe virus) to enter the insect cell, especially the cells of salivary glands [42]. *P. candelaria*, psyllid, aphid, and the small brown planthopper all belong to Homoptera insects, so our results inferred that the PyCaV virus recovered in this study might follow the route of the piercing–sucking mouthpart, the gut, the hemolymph, and the salivary gland before it is finally released into virus-sensitive plants through saliva. But this speculation needs to be supported by further determining if it has any plant hosts.

In the study, in all 123 *P. candelaria*, 33 were found carrying the PyCaV virus, and the virus detection rate (VDR) is about 26%. Compared to the 6.93% detection rate of BtAstV (Bat-transmitted astroviruses), 1.79% detection rate of BtCoV (Coronaviruses), and 0.67% of BtCalV (novel caliciviruses) from bats [43], the VDR of the PyCaV virus is relatively high, especially in the autumn and spring in FRIFAAS and the spring in Minhou County (Figure 4B), which means it is important to study its possible pathogenicity before it causes possible harm. To the different sampling regions, the VDRs varied significantly, too. The VDR is about 18% in FRIFAAS and 35% in Minhou County, which reflected the remarkable distribution difference of the PyCaV virus, which might reflect that the location or distance or other factors which can hinder the migration and spread of the insect carrier might influence the transmission of this virus (Figure 3B and Figure 4A, Appendix A). In spring, the VDR is about 46%; in summer, the VDR is about 15%; in autumn, it is about 24%; and in winter, it is about 23%. The VDR showed obvious seasonal variation, in which the VDR reaches its peak in the spring, touches its lowest valley in the summer, and rises in the autumn and winter (Figure 4B, Appendix A), and the VDR showed obvious regional variation (Figure 4A). Given the more longan lanternflies carry the PyCaV virus in the spring, it is speculated that this season may be a high incidence period of PyCaV virus infection from these insects to other possible hosts. The above results preliminarily reflected the characteristics of PyCaV virus infecting *P. candelaria* in different seasons and different locations; to deepen the role of this study in guiding pests or microbial pathogen control, the number of samples and geographical scope should be further expanded in future research.

As a new iflavirus firstly identified in *P. candelaria*, the discovery of PyCaV expands the understanding of iflavirus. In addition, since *P. candelaria* is a common fruit tree pest, whether PyCaV virus affects the pathogenicity of *P. candelaria* has important research significance. In the process of endangering fruit trees, *P. candelaria* can not only harm fruit trees by sucking fruit juice through needle-like piercing-sucking mouthparts but also cause diseases to fruit trees through their excreta [23]. The PyCaV virus is a newly discovered iflavirus from *P. candelaria*, and the hosts of iflaviruses are mostly arthropods, so it should be an entomopathogenic virus. But the insect-tissue-distribution experiments carried out in this study inferred that it might infect plants, though no solid evidence has been obtained until now. Therefore, whether the PyCaV virus can be transmitted into fruit trees by *P. candelaria*, or whether it has potential pathogenicity to infect other insects, is the future research direction. DWV, a representative iflavirus, can infect honeybees, which are the most important pollinating insect [44]. Given longan trees are important honey source plants in subtropical regions, it is worth noting whether the PyCaV virus will infect bees. The transmission mechanism of PyCaV should also be investigated in further research. Whether there is an intermediate host in the process of PyCaV infection of *P. candelaria* and whether there is co-infection with other viruses are all urgent problems to be solved.

In this study, the novel PyCaV virus carried by *P. candelaria* was identified for the first time, its distribution in the pests and the changing trends of its carrying rates by *P. candelaria* across time and space were revealed, which leads to an urgent need to explore whether this virus is harmful to longan tree or its pollinators. According to the current research, PyCaV is identified as an insect-infecting virus. The host of PyCaV, *P. candelaria*, is the focus of pest control work, especially in fruit tree production. The effect of this virus on the yield of longan and other fruit trees should be further explored. This study laid a new foundation for studying the role of *P. candelaria* as an insect vector in microbial pathogen transmission and expanded our understanding of iflavirus.

## Figures and Tables

**Figure 1 insects-15-00625-f001:**
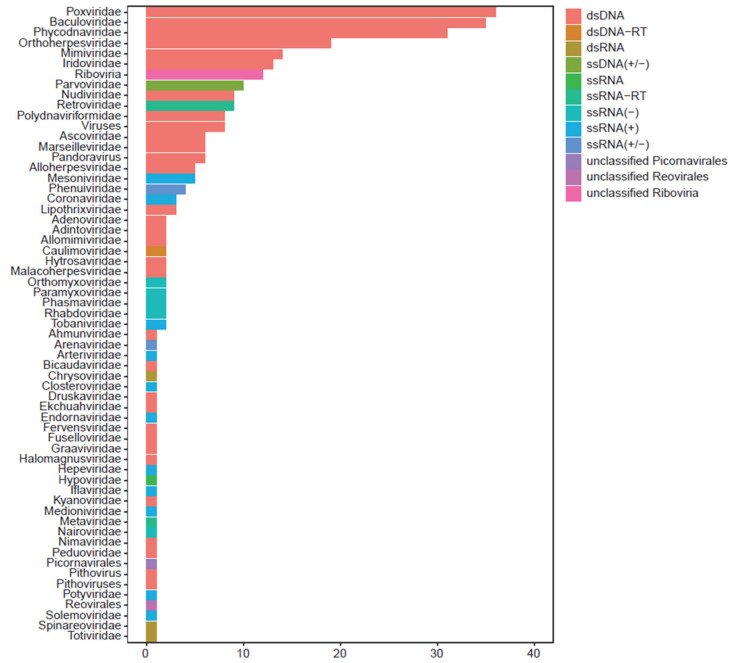
Taxonomic distribution of virus detected from *P. candelaria*. The total of 1932 viral contigs derived from 293 kinds of viruses were classified into 61 families based on ICTV and NCBI. The horizontal bar represents the number of viruses, while the color of bar represents the type of viruses in the corresponding family.

**Figure 2 insects-15-00625-f002:**
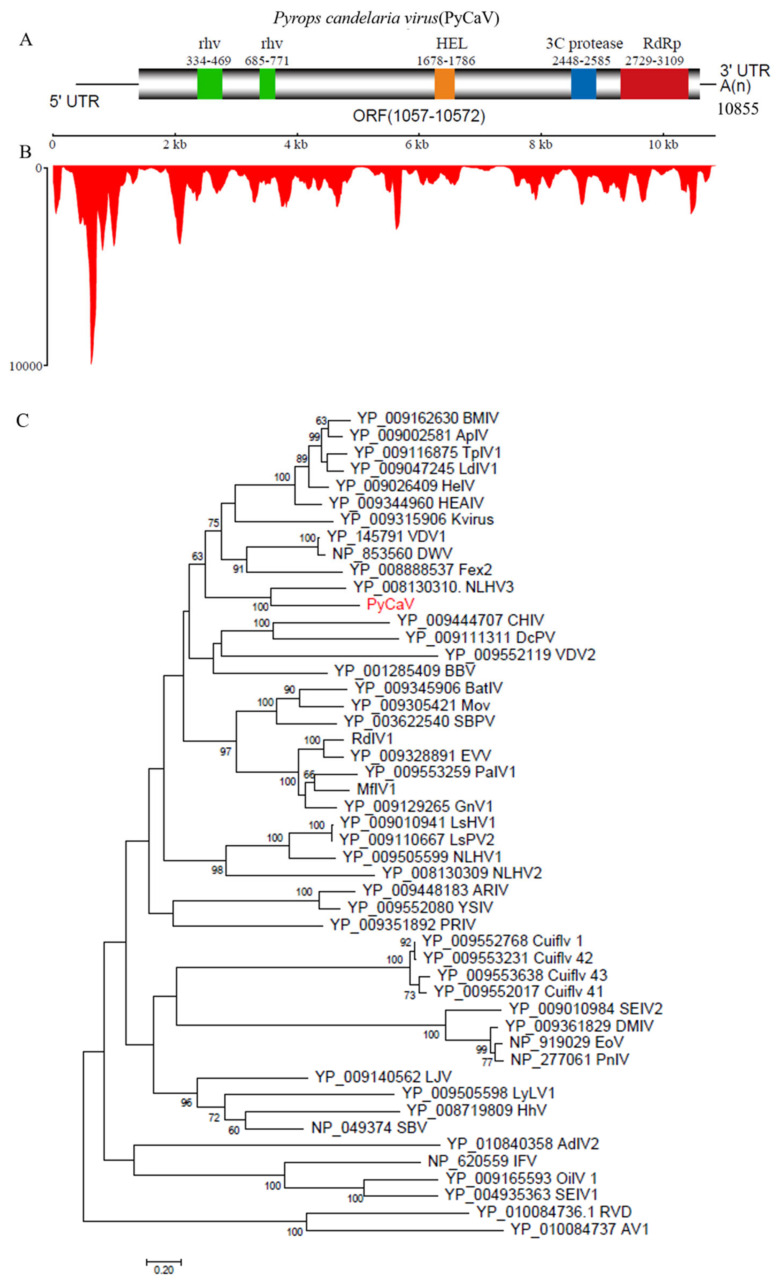
Schematic diagram of the genome structure and evolutionary tree analysis of PyCaV. (**A**) The genome encodes a 3171 aa polyprotein with several conserved domains, including two rhv domains, an RNA helicase domain (HEL), a 3C cysteine protease (Pro) and an RNA–dependent RNA polymerase (RdRp). Numbers on the bottom indicate the start and end positions of the polyprotein coding region in the genome. (**B**) Transcriptome raw read coverage of PyCaV. (**C**) Maximum-likelihood phylogenetic tree based on RdRp of PyCaV and other iflaviruses. Two members, *Rabovirus D1* (RVD) and *Aimelvirus 1* (AV1), of the family *Picornaviridae* were used as the out group. Bootstrap values (>60%) were shown at each node of the tree. Bar represents the genetic distance. Virus name abbreviations was listed in Appendix A.

**Figure 3 insects-15-00625-f003:**
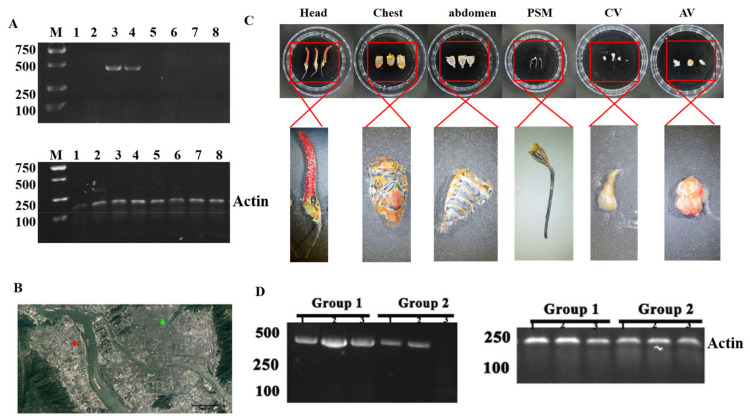
The detection of the virus in longan lanternflies. (**A**) The PyCaV was detected from the random *P. candelaria* samples. The lower panel was the amplification of actin used as the inner reference. (**B**) The map showing the sample collecting locations. The red spot is where the samples are collected in Minhou County, and the green spot is the position of FRIFAAS. (**C**) The separated tissues. The left three panels are the samples from the head, chest, and belly of longan lanternflies; the right three panels are corresponding to the piercing–sucking mouthparts from the heads, viscera in the chest, and viscera in the abdomen. (**D**) The PyCaV was detected from different parts and tissues of the *P. candelaria* samples. The left panel, Group 1: the separated head (Lane 1), chest (Lane 2), and abdomen (Lane 3) samples of longan lanternflies; Group 2: piercing–sucking mouthparts (Lane 1), viscera in the chest including salivary glands (Lane 2), and viscera in the abdomen including gonad (Lane 3). The right panel: amplification of actin as the inner reference.

**Figure 4 insects-15-00625-f004:**
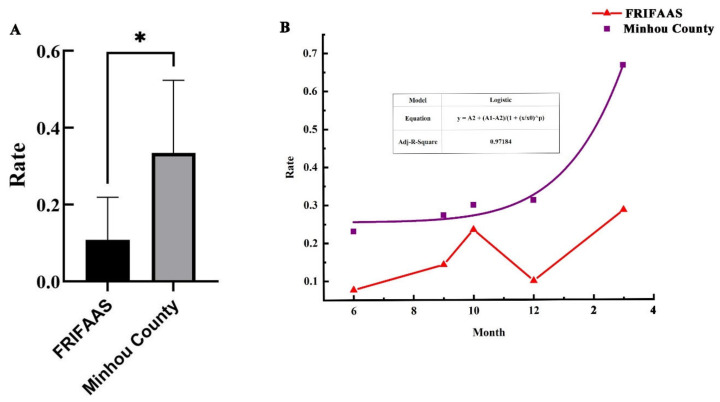
The analysis of changing trend of the PyCaV carrying rate in different sampling sites and seasons. (**A**) The statistical analysis of PyCaV carrying rate of *P. candelaria* samples from FRIFAAS and Minhou County. (**B**) The simulation of the changing trend of the PyCaV carrying rate in both FRIFAAS and Minhou County with Origin 2018. Purple squares and lines are the simulated models for samples collected from Minhou County, and red triangles and lines are the models for the samples from FRIFAAS. (* means significant difference *p* < 0.05).

## Data Availability

The original contributions presented in the study are included in the article/Appendix A. Further inquiries can be directed to the corresponding authors.

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
