# Peer review of "Screening and Characterization of a New Iflavirus Virus in the Fruit Tree Pest Pyrops candelaria"

_insects, 2024, doi:10.3390/insects15080625_

Round 1

Reviewer 1 Report

Comments and Suggestions for Authors

Dear authors

After carefully reviewing the manuscript, I have several comments and suggestions that could help improve the clarity and impact of your work.

1. The descriptions in the Materials and Methods section need to be more detailed. Please include specific information about the reagents, equipment, and primers used to ensure reproducibility. Expanding this section to provide clear and complete experimental procedures is crucial.

2. The Discussion section would benefit from a more thorough comparison with existing literature. Please provide a detailed analysis of how your findings expand or challenge current scientific understanding.

3. This section does not adequately address the limitations of the study. Acknowledging limitations, such as sample size and geographic scope, will add to the credibility of your research.

4. Future research directions are not well defined. Suggesting specific areas for further investigation, such as the mechanisms of PyCaV transmission and its impact on plants, would enhance the value of the discussion.

5. There are some grammatical and spelling errors throughout the manuscript. A thorough review to correct these issues is necessary.

6. Ensuring consistent formatting throughout the manuscript will improve its readability and professionalism.

7. All figures and tables should be clearly labeled and referenced in the text. Detailed explanations of the data presented will help readers understand your findings better.

8. The Conclusion section should succinctly summarize the key findings and their implications for pest management and agriculture. Emphasizing the practical applications of your research will highlight its significance.

Your manuscript presents a clear direction and contributes to the understanding of viral pathogens in agricultural pests. However, addressing the above points will significantly improve the clarity, depth, and impact of your study.

With Regards

Author Response

Dear reviewer,

     Thank you very much for taking the time to review this manuscript. We have studied your comments carefully and have made point-to-point revision which marked in red in the revised paper.Attached please find the revised version, which we would like to submit for your kind consideration.

     We would like to express our great appreciation to you and reviewers for these professional comments on our paper again. We are looking forward to hearing from you.

Best regards!

Yours Very Sincerely,

Zhuang Zhenhong

Reviewer 2 Report

Comments and Suggestions for Authors

As a study of a new virus in insects, this is an interesting study. The manuscript is well organized, but the discussion should focus on the results of the study, which needs to be revised and supplemented. The following questions need to be addressed.

L16: ‘The Homoptera insect’ changes to ‘The homopteran insect’.

L63: Include Hymenoptera, too.

L71: Remove ‘(S. exigua)’.

L81: Change to ‘The homopteran insect,’.

L89: Change to ‘three developmental stages:’

L112: Change ‘vectot’ to ‘vector’.

L378-426: The contents of these paragraphs are the equivalent of an Introduction, and better to move to the section of Introduction. The authors need to discuss on the results of their research.

L457-L468: The authors discuss salivary glands, insects, and plant pathogenic viruses. However, PyCaV is an entomopathogenic (insect-infecting) virus. In the case of entomopathogenic viruses, the important target organs of the insects against PyCaV are likely to be other organs rather than the salivary glands. The authors need to explain how PyCaV is pathogenic to organs and tissues of P. candelaria.

L487-492: The authors need to clarify whether PyCaV is entomopathogenic or phytopathogenic. Therefore, it is important to distinguish whether this study provides information for pest control or for understanding the function of plant virus vectors.

Author Response

(The authors gave the same response as above.)

Round 2

Reviewer 2 Report

Comments and Suggestions for Authors

The manuscript was improved for publication, but correct some minor errors in the manuscript.

L454-456: Correct 'P. Candelaria' to 'P. candelaria'.

L470: Make a space in 'P. candelaria'.

Author Response

Comment 1:L454-456: Correct 'P. Candelaria' to 'P. candelaria'.

Response 1: Thanks, we have revised it according to your professional advice.

Comment 2:L470: Make a space in 'P. candelaria'.

Response 2: Thanks, we have revised it according to your kind advice.